# The Concept of Fertility in the Field of Fruit Growing and Its Evolution from Ancient Times to Present Day

**DOI:** 10.3390/plants14182883

**Published:** 2025-09-16

**Authors:** Ettore Barone

**Affiliations:** Department of Economics, Business and Statistics (dSEAS), University of Palermo, 90128 Palermo, Italy; ettore.barone@unipa.it

**Keywords:** biennial bearing, concept review, floral biology, fruit domestication, fruit load, fruit yield, fruitfulness, *Olea europaea* L.

## Abstract

After a brief review of the concept of fertility in antiquity—from mythological, historical, religious, and artistic perspectives—this conceptual review examines the evolution of the notion of fertility in fruit growing, considering both its biological and agronomic dimensions. The discussion addresses the phenomena underlying the production process and the quantitative and qualitative yields of fruit trees, including the interactions between vegetative growth and reproductive aspects, as well as various interferences—such as alternate bearing or sterility—that mediate between potential and actual fertility. These aspects are analyzed in light of both well-established studies and the most recent research findings. Furthermore, a holistic and comprehensive approach is presented, aiming to transcend the limitations of a purely biological interpretation and to clarify certain ambiguities in the use of the term “fertility,” with particular focus on the physiology of flowering and fruiting in a paradigmatic Mediterranean fruit tree species (*Olea europaea* L. subsp. *europaea* var. *europaea*). Finally, the potential contributions of recent advances in the understanding of flowering and fruiting biology are discussed, particularly in relation to genetic improvement and the development of simulation models for the bio-agronomic behavior of fruit trees. Future perspectives are also explored, especially regarding bio-agronomic strategies to address alternate bearing.

## 1. Introduction

No other theme in horticulture—and particularly in the fruit tree sector—carries the same evocative and symbolic power as that of fertility. The term itself lends itself to multiple definitions, sometimes referring to the soil, at other times to the plant. According to the Oxford English Dictionary (accessed online in July 2025), several definitions exist for the term “fertility.” The earliest, dating back to 1490, defines it as follows: “The quality of being fertile; fecundity, fruitfulness, productiveness.” It continues by stating the following: “literally, of the soil, a region, etc.; also of plants and animals.” Over time, numerous additional meanings—both transferred and figurative—have been introduced, with contemporary usage predominantly associated with medicine and human reproduction, as noted in the *Encyclopaedia Britannica* (accessed online in July 2025).

A rough yet significant search conducted on Scopus for the term “fertility” (without year limits) returned a total of 264,425 records, the majority of which—approximately 50%—were related to “Medicine,” and around 29% to “Agriculture and Biological Sciences.” Narrowing the scope to the latter category and applying the keywords “plants OR trees OR shrubs OR vegetables” (thereby excluding “animals” and other irrelevant terms) yielded 48,523 records. Further refinement using the keywords “productivity OR yield” reduced the total to 30,065 records. Among the selected scientific papers, the most frequently represented keyword was “Soil fertility” (8659 records), followed by “Crop yield” (2374 records). This simple approximation nonetheless offers an effective snapshot of how the term “fertility” appears and is used in contemporary scientific literature, revealing a marked predominance of its application to soil-related studies (78.5%) over those focused on plants (21.5%). Still, the term “fertility” remains widely and closely linked to issues of plant productivity and yield.

This paper, following a review of the concept of fertility in antiquity—highlighting a broad spectrum of connections between mythological, historical, religious, artistic, and horticultural interpretations—aims to analyze the evolution of the concept of fertility in fruit culture, from both biological and agronomic perspectives. The goal is to highlight the specific processes that contribute to determining yield within the biennial fruiting cycle of trees, and the numerous adjustments that occur from an initial potential level of fertility to the point of harvest.

To this end, a holistic and comprehensive approach is proposed, seeking to go beyond the limitations of a strictly biological interpretation. Special emphasis is placed on the physiology of flowering and fruiting in an iconic Mediterranean fruit tree species: *Olea europaea* L. subsp. *europaea* var. *europaea*. In addition to its significance from historical, economic, social, and environmental standpoints, this species presents a range of bio-agronomic challenges related to its biological and reproductive characteristics. These serve as paradigmatic examples for exploring the fertility of Mediterranean fruit trees, as discussed in the sections that follow.

## 2. The Origins of the Concept of Fertility: Mother Earth

A first possible interpretation of how the concept of fertility was perceived in antiquity is offered by mythology. Hesiod (c. 700 BCE), an ancient Greek poet, recounts the origins of the Universe in his poem *Works and Days* (available online at https://www.sacred-texts.com/cla/hesiod/works.htm, accessed on 4 July 2025), stating the following:


*"At the time of Kronos men lived as gods […] and the fertile land spontaneously bore many and copious fruits."*


Similarly, Virgil (70–19 BCE), a Roman poet of the Augustan period, echoes this idea in his poem *Georgics* (I, 125–128), asserting that during the Golden Age, no farmers needed to work the land.

In ancient times, fertility was considered less an attribute of the plant itself and more a property of the Earth—revered as the progenitor of life and all living beings, often personified as "Mother Earth." Fruit, whose production process remained largely mysterious, was seen as a divine gift. The Romans even worshipped Pomona, the goddess who watched over fruit production.

At the dawn of agriculture, early forms of itinerant proto-agriculture became a vehicle for the spread of the first domesticated crops [1]. The expansion of Neolithic farmers from the Middle East in search of new land and fertile soil became a major driving force behind human migration [2,3].

This shift—marking the dominance of agriculture over other subsistence strategies—bound humans inextricably to the land and left an indelible trace not only in myth but also in sacred texts. An echo of this transformation is found in the Old Testament:


*“Cursed is the ground because of you;*



*through painful toil you will eat food from it*



*all the days of your life.”*
(*Genesis* 3:17, available online at https://www.bible.com/bible/compare/GEN.3.17-19, accessed on 4 July 2025)

This passage expresses both a lament for the lost Golden Age, or earthly paradise, and an awareness that the new, stable relationship with the land came at a cost—namely, the daily labor required to maintain the fertility of the soil. Yet, despite these efforts, the fruits of such labor were often disappointing or even rejected, as in the biblical story of Cain and Abel:


*“Now Abel kept flocks, and Cain worked the soil. In the course of time, Cain brought some of the fruits of the soil as an offering to the Lord. And Abel also brought an offering—fat portions from some of the firstborn of his flock. The Lord looked with favor on Abel and his offering, but on Cain and his offering he did not look with favor.”*
(*Genesis* 4:2–5, available online at https://www.bible.com/bible/111/GEN.4.2-16.NIV, accessed on 4 July 2025)

In this context, the concept of the "field" emerged—later formalized in Roman times as the Latin word *ager*, the root of the word *agriculture*. This exclusive association between fertility and soil persisted well into the Roman era. For example, rustic lands assigned to veterans were not allocated according to a fixed surface area but rather based on the estimated fertility of the soil type—a principle that later influenced modern cadastral systems.

From the 8th century CE onwards, even forests were not evaluated in terms of surface area, but according to the number of pigs that could be raised and fattened there (*silva ad saginandum porcos*—′forest for fattening pigs′). Similarly, fields were assessed based on their wheat yield, vineyards by wine production, and meadows by hay output [4].

## 3. The Symbolic Value of Fertility in the Sacred Scriptures and Arts

The sacred texts of the three Abrahamic religions—Judaism, Christianity, and Islam—frequently refer to fruits, the fertility of fields, and the productivity of plants [5]. These references often serve symbolic purposes: to celebrate the destiny of the chosen people called to inhabit the Promised Land (e.g., Isaiah 27:6), to convey prophetic metaphors (e.g., Revelation 22:2), or to deliver didactic messages promising abundance (e.g., Ezekiel 36:30).

A particularly noteworthy and environmentally grounded aspect of these texts is the link between fertility and water, especially significant given the arid landscapes of the Near East:


*“In a fertile field, along the course of great waters, it was planted, to put forth branches and bear fruit and become a magnificent vine.”*
(*Ezekiel* 17:8, available online at https://www.bible.com/bible/compare/EZK.17.8-13, accessed on 4 July 2025)

Similarly, the Qur’an evokes this connection with great poetic force:


*“We poured the water in streams, then split the earth in furrows and made grains sprout, and vines and reeds, and olive trees and palms, and thick gardens and fruit and fodder—for your benefit and your herds.”*
(*Surah 80*, available online at https://quran.com, accessed on 4 July 2025)

Fertility also acquires a symbolic value in negative contexts, such as when it becomes an instrument of destruction in warfare. For example, in *2 Kings* 3:25, fertility is deliberately undone as an act of extermination:


*“They demolished the cities; on all the fertile fields each one threw a stone and filled them; they blocked all the springs and cut down all the useful trees.”*
(available online at https://biblehub.com/2_kings/3-25.htm, accessed on 4 July 2025)

Art, in all its forms, has long engaged with the theme of fertility. Sacred painting, for instance, frequently employs fruit as a powerful symbol to convey theological meaning. The pomegranate (*Punica granatum* L.), originally a pagan emblem of fertility due to its abundant vermilion seeds, became a prominent motif in Christian iconography. It was a favored subject of Caravaggio (1571–1610) [6], symbolizing both the Passion of Christ and the spiritual fruits of redemption.

Literature, too, testifies to the evocative power of fruiting, often capturing natural phenomena with striking sensitivity. Shakespeare, for example, draws metaphorical significance from the timing of flowering and ripening. In *Othello* (II.3.376), Iago makes the following remark:


*“Though other things grow fair against the sun,*



*Yet fruits that blossom first will first be ripe.”*
(available online at https://www.online-literature.com/shakespeare/othello/7/, accessed on 4 July 2025)

This observation equates precocity in fruiting with vulnerability or early consequence—an insight drawn directly from the natural world.

In *A Midsummer Night’s Dream* (III.2.208), Shakespeare also refers to the rare natural phenomenon of fruit twinning (see Figure 1) as a metaphor for love and unity. Helena describes an inseparable bond using the image of a double cherry:


*“So we grew together,*



*Like to a double cherry, seeming parted,*



*But yet a union in partition;*



*Two lovely berries moulded on one stem.”*
(available online at https://myshakespeare.com/midsummer-nights-dream/act-3-scene-2-popup-note-index-item-double-cherry, accessed on 4 July 2025)

Even the specific theme of artificial pollination, sometimes necessary for the fertilization of dioecious species—as in the case of the date palm—is steeped in symbolic meaning. In ancient Mesopotamia, this practice was celebrated as a form of divine intervention (Figure 2) and has since served as a source of literary metaphor. A striking example can be found in the words of Ibn Zaydun (1003–1070 CE) [7]:


*“Has fecundado mi expiritu; recoge, pues, los*



*frutos primerizos.*



*Los frutos de la palmera son de quien la ha polinizado”.*
[You have fertilized my spirit; gather, then, the first fruits. The fruits of the palm tree belong to the one who pollinated it.]Ibn Zaydun (1003–1070 CE, in [7].

## 4. The Expansion and Evolution of the Concept of Fertility: From Soil Alone to Soil and Plant

Throughout history, agricultural societies have been marked by recurring cycles of abundance and famine—particularly during the 11th, 14th, and 18th centuries. Agriculture has always been called upon to meet the primary needs of the populations that depend on it. As Cicero (106–43 BCE) wrote, among all human occupations, agriculture was “the noblest, the most fruitful, the most delightful, the worthiest of a man and a free citizen.”

In stark contrast, Jared Diamond [8] presents a critical view of agriculture’s historical impact:


*“Recent discoveries suggest that the adoption of agriculture, supposedly our most decisive step toward a better life, was in many ways a catastrophe from which we have never recovered. With agriculture came the gross social and sexual inequality, the disease and despotism, that curse our existence.”*


The ambivalent legacy of agriculture—both vital and problematic—is also reflected in the worldview of some hunting and gathering societies. For instance, Smohalla, a chief of the Shahaptin people (Washington, c. 1880), offered this powerful metaphor:


*“You ask me to plow the ground! Shall I take a knife and tear my mother’s bosom? Then when I die she will not take me to her bosom to rest. You ask me to dig for stone! Shall I dig under her skin for her. Then when I die I cannot enter her body to be born again. You ask me to cut grass and make hay and sell it and be rich white men! But how dare I cut off my mother’s hair?”*
Smohalla, Chief of the Shahaptin—Indian Tribe, Washington, c. 1880. In [9].

Driven by the pressure of hunger and supported by advances in botanical science during the 16th to 18th centuries, plants were finally re-evaluated and valorized—especially key crops such as forage legumes, maize, rice, potatoes, and fruit trees. It was in this period that pomology began to emerge as a scientific discipline distinct from botany [10]. In 18th-century Europe, the modernization of agriculture was made possible through the revival of ancient crops, now integrated into more intensive and accelerated crop rotation systems [11].

From this point onward, there was growing recognition that increased productivity—and thus fertility—could be enhanced through the deliberate selection of genetically superior species and varieties. Plant improvement, once left largely to chance, became a human endeavor. Charles Darwin, in *The Origin of Species* (1859), captured this shift through his concept of “unconscious selection”:


*“The pear, though cultivated in classical times, appears, from Pliny’s description, to have been a fruit of very inferior quality.*



*I have seen great surprise expressed in horticultural works at the wonderful skill of gardeners, in having produced such splendid results from such poor materials; but the art, I cannot doubt, has been simple, and, as far as the final result is concerned, has been followed almost unconsciously.*



*It has consisted in always cultivating the best known variety, sowing its seeds, and, when a slightly better variety has chanced to appear, selecting it, and so onwards”*
(available online at https://www.vliz.be/docs/Zeecijfers/Origin_of_Species.pdf, accessed on 4 July 2025)

This marked a turning point. No longer was it sufficient to follow the advice of Roman authors like Cato the Elder (234–149 BCE), who emphasized soil management as the foundation of fertility:


*“Quid est agrum bene colere? Arare; Quid secundum? Arare; Tertio? Stercorare.”*
[What does it mean to cultivate the land well? Well plow; What is the second? Plow; Third? To manure] (De Agri Cultura, 61, 1) (available online at https://www.thelatinlibrary.com/cato/cato.agri.html, accessed on 4 July 2025)

Instead, the attention of both scientists and farmers shifted—consciously this time—towards plants and the challenge of crop productivity [12]. The concept of fertility thus re-entered the realm of genetics, millennia after its earliest expression through the domestication of useful plants. Whether this domestication is viewed as a long, unconscious process [13,14] or a relatively rapid and intentional one [15,16], the focus on genetic improvement has become central.

In the contemporary era, there is growing concern that plant breeding and agricultural innovation are increasingly driven by a criterion of unconditional productivity, sometimes at the expense of plant genetic diversity [17,18]. Valuable sources of agro-biodiversity are being abandoned—or even lost—under the pressure of high-yield demands.

At the other end of the spectrum, soil itself is sometimes reduced to the role of an inert substrate, as in the case of soilless cultivation systems, where fertility is artificially amplified in a plant–soil system that is heavily skewed in favor of the plant [19].

Since the 20th century, the impact of genetic improvement on agricultural progress—particularly through yield increases—has become undeniable [20,21]. Modern agriculture′s productivity gains are typically attributed 50% to plant breeding and 50% to improved cultivation techniques [22].

In this way, we can observe the reconciliation—at least in theory—of an ancient divergence that dates back to the first century CE, when two Roman authors held opposing views: Columella (4–70 CE) emphasized the importance of cultivation practices, while Pliny the Elder (23/24–79 CE) gave more weight to the intrinsic value of plants. Today, both soil fertility and plant fertility are recognized as equally essential.

In the sections that follow, although we acknowledge the ongoing relevance of soil fertility from multiple bio-agronomic perspectives, for the sake of brevity, we will focus on the various meanings and uses of the term “fertility” as applied specifically to the reproductive physiology and productivity of fruit trees.

## 5. The Different Meanings of Fruit Tree Fertility

### 5.1. The Link Between the Biological and Agronomic Sense of Fertility

Upon closer examination, the changes brought about by the long process of domestication [23,24] and, more recently, by the genetic improvement of fruit trees have not always led to an increase in fertility in the strict biological sense—i.e., the plant’s reproductive capacity. A recent review of reproductive trait changes associated with domestication is provided in [25]. A compelling example is that of seeds, which—viewed from the plant’s perspective—constitute a clear expression of reproductive fertility. However, under human selective pressure, seed characteristics have changed significantly over time, both in number and size, depending on the plant species and intended use.

In cereals, for instance—such as maize—both the number and size of seeds have increased compared to the presumed wild progenitor (*teosinte*), as a result of human selection [13]. Conversely, in many fruit trees, whose edible part is typically distinct from the seed, the same process has often led to an increase in fruit size but a decrease in both seed number and size. In fact, fertility in this biological sense may even disappear entirely through human selection—as in fruit species where the seed is not the edible portion. Examples include seedless (apirenic) bananas, oranges, mandarins, and grapes, where fruit development occurs through two different mechanisms: parthenocarpy and stenospermocarpy [26]. In these cases, fruit formation is independent of fertilization and/or pollination, and seeds are absent—something consumers generally find desirable.

Interestingly, seedless fruit species, having lost their ability to reproduce autonomously, become dependent on humans for their propagation and survival. As such, they have been described as a biological paradox, since they no longer contribute to the production of offspring [27]. In this context, humans have effectively reversed the course of natural evolution by redefining fertility—not as the biological ability to reproduce, but rather in agronomic terms, as productivity.

A similarly fascinating natural phenomenon that represents a modified form of sexuality is polyembryony, particularly common in citrus species [28]. This involves the development of multiple embryos within a single seed, typically due to the spontaneous formation of apomictic embryos from the maternal somatic tissues (e.g., nucellus or integuments), alongside the sexually derived zygotic embryo [29]. This phenomenon, which may involve paternal gene silencing, likely contributed to the spread of natural citrus hybrids [30]. The result is remarkable: from a single polyembryonic seed, multiple seedlings may emerge, with only one of sexual origin, while the others are diploid clones of the mother plant. This represents a rare and intriguing form of vegetative propagation through a sexual organ, namely the seed, and has long fascinated researchers due to its practical advantages, including virus sanitation.

Another major mechanism related to fertility that has been altered through human intervention is self-fertilization, which is rare in wild plants [15]. The self-incompatibility barrier—a mechanism of gametophytic sterility that evolved to prevent inbreeding—is still common in many fruit tree species, especially in pear, apple, olive, almond, cherry, and plum trees [31]. However, in modern cultivars produced through genetic improvement, self-incompatibility is being progressively reduced, offering significant advantages to growers [32]. These new cultivars are increasingly self-fertile, meaning they can self-pollinate without relying on other varieties. This reduces the risk of poor or incomplete fertilization and the consequent drop in productivity. Due to its agronomic importance, self-incompatibility—first observed in the 1700s and later studied in depth by Darwin—has been widely researched [33]. Today, overcoming self-incompatibility is a major objective in fruit tree breeding, achieved even through mutagenesis. Self-compatible cultivars have been developed in almond, cherry, and plum [31]. For example, 11 out of 21 new European plum cultivars recently developed are self-compatible (52%) [34]. Even partial self-fertility is considered beneficial, as it lowers the need for pollinators and can reduce thinning costs [35]. In contrast, in species capable of facultative parthenocarpy, maintaining incompatibility is sometimes desirable. Preventing cross-pollination ensures seedlessness—an attribute appreciated by consumers. This is achieved in *Citrus* species like clementines using agronomic methods (fine mesh nets, spatial isolation, physical barriers) or biological methods, such as the creation of sterile polyploid hybrids. One notable example is “Oroblanco,” the first triploid hybrid (*Citrus grandis* 2n × *C. paradisi* 4n), developed in California [36], followed more recently by triploid hybrids developed in Italy [37]. This trend further exemplifies the evolution of the concept of fertility, marking a growing divide between biological fertility and agronomic fertility, which no longer necessarily coincide. Even genotypes with significant biological fertility issues have been preserved due to their agronomic value. For instance, male-sterile olive cultivars such as the Italian “Cerasuola,” the Algerian “Chemlal,” and the French “Lucque” bear flowers with necrotized anthers and no trace of pollen [38]. Despite their lack of biological fertility, these cultivars have been maintained through vegetative propagation due to their high agronomic performance. On the other hand, triploid apple cultivars can produce more than eight times the usual number of pollen grains per anther. While this suggests high fertility, the viability of these pollen grains is often reduced, as is their effectiveness as pollinators.

A particularly relevant aspect of fruit tree fertility concerns reproductive barriers, such as dichogamy—either protandry or protogyny—which refers to the lack of synchrony between pollen release and stigma receptivity, as well as functional dioecy [39]. A prime example of the latter is the kiwifruit (*Actinidia chinensis*), where male plants produce viable pollen but have a rudimentary, sterile ovary, while female plants possess functional ovaries and stamens that produce degenerated pollen [40,41].

A unique expression of sexuality is also seen in carob (*Ceratonia siliqua* L.), a relatively recent domesticate. This Mediterranean species exhibits trioecy [42], or even polygamous-dioecy, as it produces male, female, and occasionally hermaphroditic flowers on different individuals. Male and female flowers arise through early selective abortion of the opposite-sex organs, whereas in hermaphroditic flowers, abortion is absent or only partial (Figure 3).

Alterations in sexual expression toward either the male or female direction in plants normally bearing hermaphroditic flowers have long been observed in several fruit tree species within the genus *Prunus* [43].

### 5.2. The Concept of Fertility as Regularity of Fruitfulness

As the preceding examples illustrate, understanding fruit tree productivity is closely linked to floral biology. Floral traits hold significant scientific and practical importance, as they directly influence both the quantity and quality of subsequent fruit production [44].

The productivity of fruit trees results from a complex, multifactorial process involving genetic, biological, physiological, environmental, and agronomic factors. It is ultimately shaped by the interplay of these factors at both the individual plant and orchard levels [45]. Unlike annual crops, perennial fruit trees are affected by long-term and interannual interactions, including competition for limited resources and correlative inhibition between vegetative and reproductive growth cycles [46,47,48,49]. These interactions often lead to irregular fruiting patterns, including the widespread phenomenon of alternate bearing (or biennial bearing)—the alternation between years of heavy yield (“on” years) and years of low or negligible production (“off” years), which occurs in both deciduous and evergreen species [50].

An insightful analysis of this phenomenon, in the context of the evolutionary transition from wild forest trees to domesticated fruit species, is offered by [51]. Although the complexities of alternate bearing cannot be explored in full here and are addressed in detail in specific studies [50,52,53], the phenomenon itself offers a valuable lens through which to reassess the concept of fertility.

If viewed solely from a biological perspective, alternate bearing may suggest an inconstant and unpredictable fertility pattern—a consequence of an inefficient or “wrong partition strategy” of the plant’s resources [54]. However, when considered from an evolutionary perspective, this behavior takes on a different meaning—particularly in tree species, where the vegetative and reproductive balance in one year greatly influences the performance in the next. Some authors interpret alternate bearing as a variant of mast seeding—a term typically applied to forest species—where “off” years facilitate a shift in resources from reproduction to vegetative growth [55]. In this model, supra-annual variation in seed and fruit output reflects a resource-based tradeoff between growth and reproduction. This tradeoff allows polycarpic trees to optimize limited resources by alternating between phases of accumulation and output [48]. Ecologically, this behavior can act as a defense mechanism: by producing an exceptionally high number of seeds in “on” years, the plant overwhelms seed predators and increases the chances of reproductive success, while the intervening “off” years suppress predator populations [56]. Though alternate bearing and mast seeding differ in key respects [15], both behaviors can be seen as ancestral strategies for natural resource management, shaped by evolutionary pressures. It is only through domestication, prolonged cultivation, genetic improvement, and the optimization of agronomic practices that humans have, in some cases, succeeded in attenuating these irregular cycles [53]. In such cases, the acquired ability for regular fruiting—which could be described as balanced fertility—is associated with enhanced regulation of fruit set. Mechanisms such as self-thinning and fruit set limitation help prevent overproduction by adjusting the final number of fruits to the tree’s physiological capacity [51]. This dynamic adjustment is a key feature of agronomic fertility and reflects a refined balance between productivity and sustainability in fruit tree self-management.

### 5.3. The Concept of Fertility as the Development of the Fruiting Process: Potential Fertility vs. Actual Fertility

Due to the polycarpic nature of fruit tree species, annual productivity is highly dependent on the proportion of bud meristems that are successfully transformed into flower buds, following a regular process of floral initiation [57]. This process, in turn, is influenced not only by environmental factors such as temperature and light (irradiation), but also by the physiological state of the tree—particularly the carbon and nutrient reserves stored in its permanent organs—as well as by the fruit load of the previous year and agronomic practices, including fertigation, pruning, and thinning.

A fundamental prerequisite for achieving desirable productivity—that is, a satisfactory tree yield—is the formation of an adequate number of fruits of acceptable size and weight. This is possible only if the tree acquires a sufficient level of reproductive activity, reflected in the successful formation of flower buds, or what is referred to as potential fertility. This potential is inherently conditioned by the inter-organ competition for available resources [49,58]. In reality, fertility in fruit trees—being the outcome of multi-year physiological processes—should be understood in reference to the various stages of the reproductive cycle: from floral induction and bud differentiation to the formation of male and female gametes, and from pollination and fertilization to the development of seeds (from ovules) and fruits (from the ovary or receptacle). However, several factors may interfere with this process. These include morpho-cytological sterility, self- and cross-incompatibility mechanisms, and progressive, dynamic self-regulation phenomena occurring at the bud, flower, and/or fruit levels. Collectively, these factors continuously adjust and modulate the initial potential fertility, ultimately determining the final fruit load and, by extension, the commercial yield, which corresponds to actual fertility [59]. Table 1 provides a schematic, though not exhaustive, overview of the flowering and reproductive characteristics of selected fruit tree species.

Different agronomic practices—such as the choice of inter-compatible varietal associations, the application of irrigation, fertilizers, and plant growth regulators (PGRs), as well as pruning techniques including shoot bending, girdling, scoring, and fruit thinning, and the improvement of the tree’s overall trophic conditions—can all contribute to increasing or, when needed, limiting fertility, thereby ensuring the commercial success of fruit orchards.

In this context, the term *fertility* is widely used in viticulture, often in relation to vine buds, or for the calculation of a fertility index (e.g., number of bunches per shoot) [60].

A significant body of scientific literature has long been dedicated to key aspects of the fruiting process, including floral initiation [47,63], flowering [31], fruit set and development [54,61], flower and fruit abortion [64], reproductive strategies [59], biennial bearing [65], and thinning [66].

Ultimately, plant productivity—i.e., *actual fertility*—represents the final outcome of a long chain of interrelated physiological events, which take place both at the level of individual reproductive organs and at the whole-plant level. These processes can only be fully understood and assessed through a holistic approach.

## 6. Towards a Unified Concept of Fertility

### 6.1. Reconciling Biology and Agronomy

As the previous sections have shown, defining plant fertility solely in terms of biological reproductive capacity is inadequate for describing the fruiting process as it relates to the agronomic productivity of fruit tree species. Therefore, it is useful to work towards a unified concept of plant fertility, framed within a broader systemic context—one that, while focused on plant productivity, intentionally excludes other common uses of the term *fertility*, such as that of soil fertility. To this end, it is first necessary to disaggregate the various components that contribute to fertility in the agronomic sense—i.e., to the productivity or fruitfulness of the plant. For simplicity, this analysis will exclude orchard-level factors (e.g., planting density, cultivation systems) and instead focus solely on the “plant system.”

The model plant used for this discussion is the olive tree (*Olea europaea* L. subsp. *europaea* var. *europaea*), a species known for exhibiting several factors that disrupt regular fruiting, such as alternate bearing, self-sterility, female or male sterility, and parthenocarpic fruit set. Notably, the olive tree is also universally recognized as a symbol of fertility, as well as of peace, wealth, and fame [14].

The flower is generally considered the obligatory starting point of the production process, as it represents the morphogenetic basis of fruit development. In the olive tree, each inflorescence bears a variable number of flowers (typically 10–35, depending on the cultivar), which may be either perfect (hermaphroditic) or staminate (male) (Figure 4). Only the perfect flowers are capable of setting fruit. These are most commonly found at the tips and primary branches of the inflorescence, while the secondary branches tend to bear staminate flowers, which typically bloom later [38,67].

The inflorescence is generally considered a unit of fruitfulness in olive trees, as many cultivars typically produce only one fruit per inflorescence [68]. At the other end of the process, the final yield per tree is determined by two key factors: the number of fruits harvested and their individual weight. In a given year (n), the number of fruits is the result of the plant’s ability to retain a certain proportion of the fruits formed after the fertilization of perfect flowers. In olives, this retention rate can be extremely low, sometimes as little as 20%, depending on factors such as cultivar, crop load, growing conditions, position on the tree, and flower position within the inflorescence [69] and references therein. The weight of individual fruits is influenced by the resources available during the fruit development phase. These resources are limited both by the fruit load (i.e., the number of developing fruits) and by the availability of reserves and newly synthesized assimilates, which depend on factors such as leaf area, leaf number, nutritional status, and water availability. Notably, both the leaf surface area and the number of perfect flowers per inflorescence are largely determined during the year preceding fruiting [69]. Therefore, the photosynthetic activity and fruit load of year n-1 directly influence the number and length of shoots that will bear a variable number of buds, depending on the induction and floral differentiation processes governed by promoting or inhibiting stimuli received by the tree [43].

The maximum productive potential—referred to as the potential fertility or yield potential—of a tree is typically reached just before anthesis. In contrast, the post-anthesis period determines the limitations on actual production, i.e., actual fertility [70]. For example, it is well documented that olive trees with a higher number of flowers at anthesis tend to exhibit a greater percentage of aborted flowers and ovaries [71,72]. Thus, actual fertility is heavily conditioned by internal competition phenomena within the plant, which result in a progressive reduction in fruit load.

In olive trees, this adjustment process mainly takes place in the 35–40 days following full bloom, involving shedding of imperfect (staminate) flowers, drop of perfect unfertilized and fertilized flowers, parthenocarpic fruit drop, and abscission of normally developing fruits at various stages. The close inverse relationship between crop load and fruit size is well established in fruit trees [73], as is the inverse relationship between crop load and vegetative activity, and between crop load and both fruit ripening and oil quality in olives [74,75,76]. In fact, crop load itself acts as a self-regulating mechanism in the olive tree by influencing the carbon balance and directing assimilates between vegetative and reproductive growth. This makes it one of the primary determinants of the tree’s productive potential. As a result, olive trees with high fruit loads often exhibit reduced leaf area and limited shoot development, while the opposite is true for trees with light crop loads [77,78,79,80]. A schematic representation of the fruit tree production cycle is provided in Figure 1.

As shown in the square box in Figure 1, growing knowledge of olive flower biology and fruiting physiology has led to integrated interpretative frameworks that emphasize the role of individual floral components [81]. Increasing attention is now paid to genetic aspects—previously underexplored or studied in isolation—such as floral structure, organ development, and their potential influence on fruit number, fruit size, and overall quality. Within this broader and more integrated concept of *flower quality*, key parameters now include starch reserves in pistils, ovary size, ovule development and longevity, and pollen viability [62,71,81,82,83,84,85].

Moreover, the self-regulating capacity of the tree—at various stages of the fruiting cycle—is crucial to ensuring fruiting constancy and regularity. The earlier a load-regulation phenomenon occurs, the more it can be compensated for by subsequent adjustments. For example, in olives, a high rate of ovary abortion may be offset by greater fruit set [86], which in turn may result in increased fruit drop due to resource competition. Thus, studies that focus solely on one early parameter—such as ovary abortion or pollenizer performance—may be incomplete or misleading when attempting to assess the actual productive capacity of a cultivar. In contrast, species with lower self-regulatory capacity are more prone to marked yield fluctuations, as seen in trees with strong biennial bearing tendencies. In olive, the initial biological investment in reproductive structures is substantial: during an "on" year, a single tree may produce up to 500,000 flowers. Yet only 1–2% of these reach maturity, which is still generally sufficient to ensure abundant and regular productivity [87]. This large flowering, despite low fruit set, has prompted various interpretations. One explanation emphasizes the high proportion of staminate flowers and their role in enhancing male fitness. This strategy may serve to maximize pollen dispersal, ensuring genetic contribution to the next generation at lower reproductive cost [88,89]. Sutherland [61] further hypothesizes that low fruit set in self-incompatible plants results from selective abortion and represents a “bet-hedging” strategy. By contrast, species such as cherry (*Prunus avium* L.), with a higher fruit set rate (25–30%), produce fewer flowers than olives. The olive’s strategy, evolved in harsh environments, is thus one of cautious redundancy: a high initial number of reproductive structures increases the system’s reliability, allowing for stepwise fertility adjustments in response to fluctuating environmental conditions over a fruiting cycle that may last a year or more [90]. Such redundancy is not limited to flowers—it also appears in the ovary, which contains four ovules, of which typically only one is fertilized [68]. Importantly, while flowers are numerous, their energetic cost is much lower than that of fruits, especially once the stone (endocarp) begins to sclerify. From a potential fertility standpoint, the tree makes a significant initial investment—referred to as reproductive effort (*sensu* [91])—whose final outcome is uncertain and subject to internal and external modulation.

This reproductive strategy resembles a principle articulated by Carl von Clausewitz (1780–1831), the Prussian general and strategist, regarding numerical superiority in battle: “*The superiority in numbers is the most important factor in the result of a combat, only it must be sufficiently great to be a counterpoise to all the other co-operating circumstances*.” (available online at https://clausewitzstudies.org/readings/OnWar1873. Accessed on 4 July 2025).

This dynamic process of ongoing fruit-load adjustment explains how even cultivars with significant fertility disorders (e.g., the Italian olive cultivar Morchiaio, with >80% pistil abortion) can still achieve good yields. It also clarifies the inconsistency of early interventions (e.g., thinning or pollenizer trials), as internal compensatory mechanisms may neutralize or override these efforts [68,92]. A clear example of such compensation is the reduction in the number of perfect flowers per inflorescence induced by thinning, which is often offset by an increase in fruit set per flower. Similarly, small-fruited olive cultivars typically produce more fruits than large-fruited ones [93].

All of these compensatory phenomena reflect the evolutionary reproductive strategy of the olive tree [62]. Within this complex and adaptive reproductive system, it seems appropriate to introduce a new definition of final fertility: Compensated Bio-Agronomic Fertility (as illustrated in Figure 1). This term defines final fertility as the result of interactions between biological and agronomic factors across the entire fruiting cycle—offering a realistic and integrated understanding of fruit yield.

### 6.2. A Digression to Broaden Our Vision

In this context, it is intriguing—though speculative—to consider a teleonomic analogy between plant fertility strategies and those found in the animal kingdom, where fertility control also occurs at various organizational levels (individual, group, species). In both cases, environmental constraints, resource availability, and—in humans—social and cultural factors play a role in modulating reproductive output. In many deciduous fruit trees, floral induction for the following year takes place during the summer–autumn period, when current-year fruit development is still ongoing [31]. Thus, the plant faces a dual challenge: completing the current fruiting cycle while simultaneously initiating the next. Often, this conflict is resolved in favor of current production, to the detriment of future fertility. This prioritization is typically regulated via hormonal inhibition, such as the production of gibberellins by the developing embryo—or even by the ovary as a whole—that suppresses floral induction [47,53,94]. Analogous processes in the animal world include amenorrhea during lactation, allowing mothers to focus on current offspring, and spontaneous abortion, practiced in some traditional societies, to protect dependent children. Similarly, infanticide (in some hunter-gatherer cultures), linked to extreme resource scarcity, can be considered analogous to natural fruit drop, reducing resource competition. Even abstinence, historically used to space births and regulate fertility, recalls alternate bearing in fruit trees. These biological and evolutionary mechanisms are all present in the olive tree and, as such, make its reproductive behavior a paradigmatic case study for broader analyses of fertility.

## 7. Concluding Remarks

This article aimed to trace the evolution of the concept of fertility over time, along with its multiple and sometimes divergent meanings. Today, despite its widespread use, the term *fertility*—when referring to the reproductive capacity of fruit tree species—remains ambiguous. As previously discussed, only at the end of the fruiting cycle can the actual yield be equated with final, effective fertility—what this paper defines as “compensated bio-agronomic fertility”. Special attention has been given to the olive tree, a species that offers multiple examples of irregular fertility expression. Several factors have historically hindered its genetic improvement, including a long juvenile phase, asexual propagation, incompatibility phenomena, and the prevalence of numerous locally adapted cultivars. Indeed, it is well known that many currently cultivated olive varieties have an unknown pedigree [95] and have undergone little to no genetic improvement since their ancient domestication [96,97]. Some may be only a few generations far from wild clones [14], highlighting the intrinsic value of the olive as a true "plant of civilization" and, as it has been aptly described, a living archeological asset [98]. On the other hand, it is increasingly clear that the traditional parameters used to define *potential fertility* are no longer sufficient. Instead, a broader and more nuanced concept of "flower quality" must be adopted [85]. Recent research on flowering genes, particularly FT (FLOWERING LOCUS T), appears promising in clarifying the hormonal mechanisms underlying floral induction [99] and may finally illuminate one of the most elusive processes in fruit tree biology. If, as hoped, such studies lead to a deeper understanding of the complex biological mechanisms underpinning fruiting in perennial species, then more precise and effective agronomic interventions may follow—ultimately enhancing fertility. One of the key future challenges is the phenomenon of fruiting alternation. Much is expected from ongoing advances in our understanding of flowering and fruiting processes and their integration into fruit tree breeding programs aimed at improving the self-regulating capabilities of fruit trees. This improved knowledge will allow for the targeted identification of bio-agronomic traits—alongside architectural and vegetative characteristics [100]—useful for guiding both traditional and innovative breeding strategies, including efforts to shorten the juvenile phase [101]. Finally, it is expected that these insights will also provide valuable input for the development of crop simulation models [102], which are widely regarded as essential tools for addressing the future challenges facing the horticultural sector.

## Data Availability

The data supporting the reported findings regarding the recurrence of the term “fertility” in the scientific literature were obtained by carrying out a search on Scopus (Elsevier, 2025) using the following search terms and iterative procedure: (1) TITLE-ABS-KEY (fertility); (2) TITLE-ABS-KEY (fertility) and (LIMIT-TO (subjarea,”agri”)); (3) (TITLE-ABS-KEY (fertility)) and (plant or tree or vegetable) and (LIMIT-TO (subjarea, “agri”)); (4) (TITLE-ABS-KEY (fertility)) and (plant or tree or vegetable or shrub) and (productivity or yield) and (LIMIT-TO (subjarea,”agri”)) (accessed on 4 July 2025).

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
