# Peer review of "The Concept of Fertility in the Field of Fruit Growing and Its Evolution from Ancient Times to Present Day"

_plants, 2025, doi:10.3390/plants14182883_

Round 1

Reviewer 1 Report

Comments and Suggestions for Authors

The manuscript referenced as plants-3801685 and titled The Concept of Fertility in the Field of Fruit Growing: A Journey Through Myth, History and Agronomy is focused on the concept “Fertility” and its evolution from ancient times to the present day. It highlights the importance of this term and its meanings in religion, the arts, botany, and agronomy.

This manuscript has involved the author in an exhaustive review of robust and recent scientific work, which I consider to be a well-argued and novel review. The work is well structured and sequenced, making it easy to read.

In my opinion, this review should be accepted for publication in Plants. It is perfectly within the scope of the journal and undoubtedly contributes to clearing up many doubts for scientists who conduct research in the field of Crop Production.

Reviewer 2 Report

Comments and Suggestions for Authors

Manuscript ID plants-3801685

Type Review

Title: The Concept of Fertility in the Field of Fruit Growing: A Journey Through Myth, History and Agronomy

Authors: Ettore Barone

Overall assessment. The paper offers an extensive and original analysis of the concept of fertility in fruit growing, starting from its mythological and historical origins and reaching contemporary biological and agronomic perspectives. The extensive documentation and interdisciplinarity are to be appreciated. The article links biological sciences, agronomy and cultural elements. The author uses the olive tree (Olea europaea) as a model species to exemplify the complex interactions between biological, potential, real and agronomic fertility, integrating data from the literature, historical observations and new conceptual perspectives, such as the proposed definition of “compensated bio-agronomic fertility”.

Strengths:

  • The text is organized into thematic sections (introduction, history, conceptual evolution, biological/agronomic aspects, proposals for unifying the concept).
  • The paper cites relevant sources, both classical and contemporary, with a solid interdisciplinary coverage.
  • The approach combines biology, agronomy, history, cultural symbolism and even art, which can be attractive to a wide audience. Extensive documentation - covers modern and classical sources.

Weaknesses:

  • Lack of a methodology section.
  • Overly extended narrative structure, too many cultural details, too long passages that can diminish the scientific part.
  • Abstract too descriptive.
  • Lack of summary tables/figures (comparative tables of fertility types, influencing factors and implications).
  • Dispersed conclusions - there is no clear final section of "Conclusions and Future Perspectives". Although there are conclusions integrated into the text, a separate final section would be useful in which the main ideas, gaps in the literature and future research directions would be synthesized.

Recommendations:

  • Include a methodology section.
  • In the Abstract it would be good to clearly specify: the purpose, methodology, main synthesized results and possible conclusions/future perspectives.
  • In the Introduction section, the objectives and research questions of this review should be clearly specified; Justification for choosing Olea europaea as a model species
  • Although interesting, some historical-cultural-literary sections can be condensed to keep the focus on the scientific analysis.
  • There are a few figures in the paper, but it would be useful to add a comparative table + conceptual diagram with the types of fertility and the factors that influence them (genetic, physiological, environmental, technological) or a generalized scheme of the process from potential fertility to real fertility.
  • Restructuring the conclusions.
  • The study of a more extensive bibliography would have been more convincing in supporting the statements made by the author. Introduction of more recent references (2018-2025), especially in the sections related to: genetic and biotechnological advances in fruit growing; recent studies in physiology and agronomic management; analyses and reviews from recent years.
  • I recommend a more concise and scientific title.
  • The sections of the paper could be better structured and organized.

The paper requires significant restructuring. Improvements (revisions) are needed in both the structure, methodology and graphic presentation, as well as in the synthesis and formatting. 

Reviewer 3 Report

Comments and Suggestions for Authors

Some of the historic parts of the manuscript are not connected to fertility of fruit trees, but basically with the idea of agriculture in general. Parts of the manuscript could be shortened or removed, especially ones not connected to fertility.

L60, L196 unclear sentence, please improve

L91 connect the two better to clarify the meaning of the sentence

L115 ‘toil’ is correct word and what is its meaning?

L201 latin name in italics

L615 what is ‘bet hedging’? Explain, please

Reviewer 4 Report

Comments and Suggestions for Authors

This is an interesting, easy to read, and very well written review.

It seems a bit unusual to have a review that starts with a general and broad concept of “fertility” in ancient literature and ends up with detailed discussion on a specific biological sense of fertility in fruit trees and olive in particular. It seems a bit too broad initially and too narrow at the end. But if this is ok with the editor and the journal style and topic, then perhaps this is ok. Except for the fact that some of the ancient texts reported and discussed, do not seem to really deal with fertility, but simply with agricultural production in general. Perhaps some of the less relevant and less related parts could be omitted, which would also partially solve the unusually large approach from prehistory to detailed discussion on restricted aspects on a single species.

Another general issue: The initial concept of fertility concerns mainly soil fertility, while all the discussion on current science focusses on biological plant fertility (in terms of flower quality, fruit set etc.): a very different concept. It is true that this second concept of fertility is more modern and was less understood or absent in ancient literature, but that is not to say that soil fertility is no longer a valid concept today. The way the review is organized and written seems to indicate that intending fertility as a property of the soil was an ancient way of looking at it, while a modern way is to look at it is as a biological plant property. As if soil fertility is no longer important, or even real, being replaced by a modern concept of fertility based on plant biological properties. In fact, soil fertility is still very modern, real, and important, while plant biological fertility is simply something else, which has been studied and understood more recently. In fact, if the focus of the review is the latter, and ancient literature did not address these concepts of plant biological fertility, then it seems out of scope to discuss the former (i.e. soil fertility) in antiquity in this review. Otherwise, soil fertility should be addressed also in moder times (as soil fertility indeed allows the plant to reach its potential and actual biological fertility). But this would make the review too wide and long. So perhaps the review should start from as far back as the literature goes on plant biological fertility, omitting the historical excursus on soil fertility, which, as mentioned, also makes the review too broad and inconsistent.

Aside from these general problems with the general approach of the review, here are few more detailed suggestions concerning the modern part (plan biological fertility).

The concept of Compensated bio-agronomic fertility appears in the figure, with no mention in the text, except for in the conclusion. I would suggest omitting this unclear and unclarified definition. But if not, then explain it clearly in the discussion, before recalling it in the conclusion.

Additionally, in the figure, the phenomena leading to “fruit weight” overlap partially with those of fruit no. It is confusing to repeat them: leaf area and No. of leaves should be reported only once, and connected to both fruit No. and Fruit weight. Furthermore, Fruit weight also affects fruit number and vice versa, therefore there should be arrows connecting them. Otherwise, the graph does not represent the reality described in the paper.

In the conclusion, you correctly write:

“The improved knowledge of olive tree flower biology and fruiting physiology has led, in fact, to recent approaches to the problem which tend to emphasize the role of the individual components of the flower under a single interpretative framework [94].”

Again, this comes up only in the conclusion and not in the discussion. This wholistic interpretation should be addressed in the discussion, and then summarized in the conclusion. Lines 573-602, could be a good place to introduce this, since that is where this topic is discussed and it is concluded that working on few aspects is “incomplete and misleading” which is exactly the conclusion reported in 94 and previous work cited therein.

In lines 578-582 and important phenomenon is described, but only old literature is mentioned (77 and 78). There is more recent and updated literature on olive tree architecture and how fruit load affects it, for instance in Tree Physiology 2018 and in Frontiers 2024.

The text in lines 612-614 discusses theories for the abundant flowering of olive. But the flowing text (lines 614-644) addresses only partially these theories, and concludes that abundant flowering is necessary to achieve good fruit set, which is in contrast with the theories cited above, explaining that that the abundance of flowering, in the face of low fruit set, is not a waste or resource or an extremely conservative approach, but is an investment in male fitness, thus not a waste from the plant fitness perspective.

It is suggested to either omit this part (i.e. line 614-644) and leaves it with the citations reported (81-82), or briefly explain all the theories, without concluding (with the Von Clausewitx example) that high flower number is important to guarantee the result (i.e. high fruit load), which is not the case, as all theories cited (81, 82) agree.

Line 655: Again, in addition to 87, you could cite more recent literature, like 94.

Reviewer 5 Report

Comments and Suggestions for Authors

Dear author,

I have read the manuscript under title `The Concept of Fertility in the Field of Fruit Growing: A Journey Through Myth, History and Agronomy `, written by Ettore Barone. This is an interesting topic with important facts for everybody who works with fruits or being a breeder, and honestly, I have enjoyed reading it, but I wonder if this is science or just historic/social/evolutional review (especially first part) of the word `fertility`? Lots of effort was used to compile such a review.  Maybe, this kind of journal is not the right place to publish it.

If the editor decides to publish it, the reorganization of the manuscript is needed in order to look like a `natural science` publication.

Here are some things that should be corrected:

  1. The title should have Olear europea, because the manuscript deals with the olive tree in most case.
  2. Abstract - three times `of` in the first line. Rewrite the sentence.
  3. Line 19 - “Fertility” should be with the small letter, here and in the whole manuscript
  4. Key words – exclude `concept review`, change `crop domestication ` to `fruit domestication` and `crop yield` to `fruit yield`; exclude Olea europea.
  5. Lines 28-29 – “And so it goes: literal of the soil, a region, etc.; also of plants and animals”. This is unclear.
  6. If the first whole paragraph is copied from Encyclopedia Britannica, then this should be reference no.1.
  7. In the aim of the manuscript you wrote that you are referring mostly to Olea europea. While reading it, I have realized that many other fruit species were mentioned, so I would rephrase the aim. You should exclude Olea form it.
  8. Since this should be a scientific review, I encourage the author to exclude all the verses, and just write about who, when and where mentioned some important fact like ` fertility`. With all the verses we are loosing scientific sound .
  9. All Latin words must be in italic.
  10. All the stories about Hesiod, Virgil, Pliny the Elder, Xenophon, Cicero and all others mentioned in the text are overtook from some literature. This should be listed as references.
  11. Paragraphs 2.0, 3.0 and 4.0 should be cut to half. Please, just mention facts.
  12. In some parts of the Sections 5.0 and 6.0 personal view of the author regarding fertility of fruits is written. This should be excluded, because this is a review of what has been done so far regarding fruit fertility. Besides, a table about different types of pollination, fertility, self-compatibility,..... in fruits should be made.  
  13. Conclusions should be half of the page, maximum. Now it is too long.

Round 2

Reviewer 5 Report

Comments and Suggestions for Authors

Dear author,

some things are corrected but many suggestions were not adopted.

But if we skip different opinions about title, aim, verses and personal opinions, I insist that points 10 and 11 (from my first review) must be fulfilled.

Also, a table from the point 12 must be added. Conclusion is still too long.
